# Accelerating Inference of Retrieval-Augmented Generation via Sparse Context Selection

Yun Zhu[1], Jia-Chen Gu[3], Caitlin Sikora[2], Ho Ko[2], Yinxiao Liu[1], Chu-Cheng Lin[2], Lei Shu[1], Liangchen Luo[1], Lei Meng[1], Bang Liu[4], Jindong Chen[1]

[1]Google DeepMind
[2]Google
[3]University of California, Los Angeles
[4]Université de Montréal & Mila

{yunzhu,csikora,hoko,canoee,kitsing}@google.com
{leishu,luolc,leimeng,jdchen}@google.com
gujc@ucla.edu,bang.liu@umontreal.ca

## ABSTRACT

Large language models (LLMs) augmented with retrieval exhibit robust performance and extensive versatility by incorporating external contexts. However, the input length grows linearly in the number of retrieved documents, causing a dramatic increase in latency. In this paper, we propose a novel paradigm named Sparse RAG, which seeks to cut computation costs through sparsity. Specifically, Sparse RAG encodes retrieved documents in parallel, which eliminates latency introduced by long-range attention of retrieved documents. Then, LLMs selectively decode the output by only attending to highly relevant caches auto-regressively, which are chosen via prompting LLMs with special control tokens. It is notable that Sparse RAG combines the assessment of each individual document and the generation of the response into a single process. The designed sparse mechanism in a RAG system can facilitate the reduction of the number of documents loaded during decoding for accelerating the inference of the RAG system. Additionally, filtering out undesirable contexts enhances the model's focus on relevant context, inherently improving its generation quality. Evaluation results on four datasets show that Sparse RAG can be used to strike an optimal balance between generation quality and computational efficiency, demonstrating its generalizability across tasks.

## 1 INTRODUCTION

Large language models (LLMs) have attracted increasing attention and exhibited impressive abilities to understand instructions and generate fluent outputs in natural language (Brown et al., 2020; Ouyang et al., 2022; Touvron et al., 2023; Team et al., 2023). Nevertheless, LLMs inevitably manifest hallucinations (Ji et al., 2023) due to their struggle with factual errors and inability to secure the accuracy of generated text solely by the parametric knowledge they encapsulate (Zhang et al., 2023; Muhlgay et al., 2024). Feeding the source of truth to LLMs in the format of retrieved context segments (Reid et al., 2024) alleviates this problem. The technique is widely known as Retrieval-Augmented Generation (RAG) (Lewis et al., 2020b; Li et al., 2022; Guu et al., 2020a).

Although the RAG framework is empirically shown to be effective, it can be expensive to scale up. This is because it requires prepending relevant documents retrieved from an external knowledge corpus to the queries (Guu et al., 2020a). As a result, the input length grows linearly in the number of documents, causing a dramatic increase in latency when using a standard Transformer whose latency scales quadratically with the input length. Some prior works such as Fusion-in-Decoder (FiD) (Izacard & Grave, 2021) and Parallel Context Windows (PCW) (Ratner et al., 2023) have proposed to alleviate this issue. Yet these methods fail to strike an optimal balance between generation quality and computational efficiency. FiD was originally designed for the encoder-decoder architecture, and thus is not compatible with currently prevalent decoder-only architectures without significant changes.

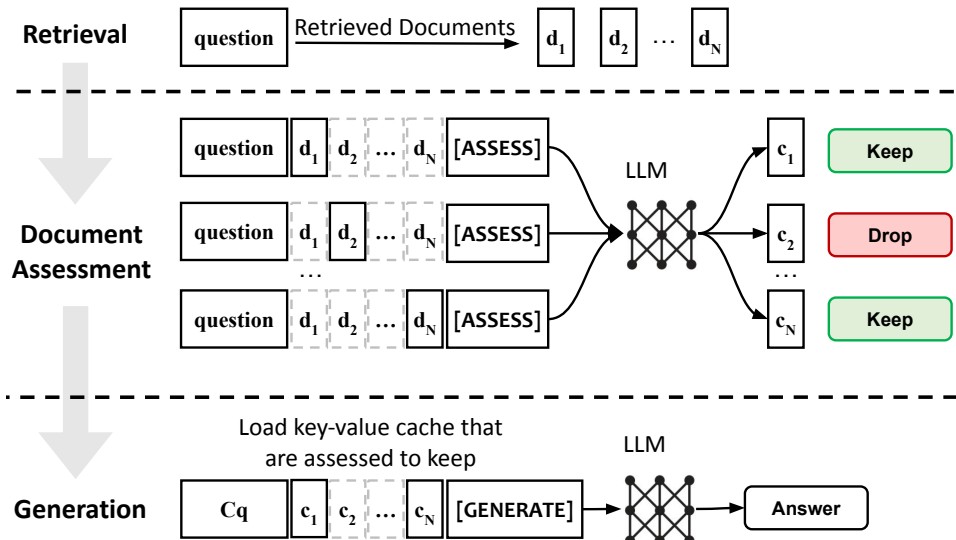

Figure 1: An overview of Sparse RAG at inference. Each of the retrieved documents $d_i$ is assessed for relevance by the LLM and irrelevant documents are dropped. Then, the KV caches $c_i$ for the remaining documents are used for generation.

While PCW can be applied to decoder-only LLMs, it only speeds up the model pre-filling and still incurs high latency since the whole context window cache is still being attended to when decoding each token. Moreover, the heavy reliance of generation on the retrieved knowledge raises significant concerns about the model's behavior and performance in scenarios where retrieval may fail or return inaccurate results (Shi et al., 2023). A typical approach for mitigating this issue is to rely on an external classifier to rank or filter the documents before prepending them to the input (Yan et al., 2024), but this process requires extra model calls which adds new complexity to inference.

In light of the issues above, we propose a novel paradigm called Sparse RAG. It operationalizes through massive pre-filling, where the key-value cache is generated by a single forward pass of the input tokens, and selective decoding, where the output is generated by attending to only highly relevant tokens auto-regressively. Previous works where the length of the retrieved contexts during pre-filling are equal to that during decoding are called *dense*-RAG in this paper. Sparse RAG, on the other hand, causes the decoding context to be significantly shorter than the pre-filled context, where retrieved documents that are not highly relevant to the input query have been dynamically dropped. Furthermore, Sparse RAG combines the assessment of each individual context and the generation of the response into a single process, in which special control tokens are used to prompt the LLM to assess the relevance of each retrieved context, and then only the key-value caches of the most relevant contexts are loaded for decoding using another control token.

The design of Sparse RAG has two additional unique advantages. First, by reducing the number of key-value cache loads during the decoding process, the LLM can achieve lower latency where it is typically constrained by memory usage. Second, filtering out undesirable contexts enhances the model's focus on relevant contexts, inherently improving the quality of the generated output. To demonstrate the effectiveness and efficiency of the proposed method, we evaluate on four datasets: PopQA (Mallen et al., 2023), QMSum (Min et al., 2023), TriviaQA (Joshi et al., 2017), and HotpotQA (Yang et al., 2018). Experimental results show that Sparse RAG can achieve similar or better quality and much better latency compared with standard *dense*-RAG or PCW-RAG approaches. Moreover, the choice of the four datasets, which include short- and long-form generation, question answering, summarization, and multi-hop reasoning, demonstrates the generalizability of the Sparse RAG approach.

## 2 RELATED WORK

**Retrieval-Augmented Generation** RAG is a family of techniques for generating output while using retrieved nearest-neighbor context data as a reference. It typically involves two stages: retrieval and generation. Retrieval finds most similar contexts based on BM25 or learned embeddings, where

Table 1: Comparisons with existing RAG-related works.

| Approach | Corrective | No extra model | Prefill efficiency | Decode efficiency |
|---|---|---|---|---|
| RAG (Lewis et al., 2020b) | No | Yes | No | No |
| Corrective RAG (Yan et al., 2024) | Yes | No | No | No |
| PCW RAG (Ratner et al., 2023) | No | Yes | Yes | No |
| Sparse RAG (Ours) | Yes | Yes | Yes | Yes |

the context can be represented as token embeddings (Khandelwal et al., 2020; Yogatama et al., 2021), dense embeddings (de Jong et al., 2022) or raw text (Guu et al., 2020b; Izacard & Grave, 2021; Lewis et al., 2020b). Once those contexts are retrieved, different architectures are leveraged to incorporate them into the model. Popular approaches include concatenation (Izacard & Grave, 2021; Lewis et al., 2020b) and cross-attention (Borgeaud et al., 2022; Lewis et al., 2020a).

In recent years, LLM architectures have evolved towards decoder-only models with significantly larger sizes. To this end, concatenation of raw text (Lewis et al., 2020b) is becoming popular for its simplicity and practicality, and many advanced approaches have been developed on top of it. Yoran et al. (2024) designed an NLI model to identify irrelevant contexts and improve robustness. Jiang et al. (2023b) actively anticipate future content and decide when and what to retrieve in long-form generation. Self-RAG (Asai et al., 2024) is proposed to selectively retrieve knowledge on an as-needed basis, by introducing a separate critic model. The critic model generates "reflection" tokens to indicate whether to retrieve information. It runs inference on each document once and uses additional "reflection" tokens to select excerpts from the documents to use for generating the response. In contrast, we unify the generation of the special control tokens and regular vocabulary tokens with one single model, eliminating the additional model and computational overhead. CRAG (Yan et al., 2024) explores and designs corrective strategies for RAG to improve its robustness of generation. Specifically, an external T5 model is trained and used to determine the usefulness of the retrieved context. Generally, these approaches explore retrieval as a useful tool to augment generation and whether retrieval is necessary.

**Efficiency in RAG** The efficiency of LLM inference is a widely explored research area, where different categories of approaches have been studied, often targetting LLM inference in general rather than RAG specifically. Some works focus on architecture-level acceleration; examples include efficient attention (Shazeer, 2019), Mixture of Experts (Fedus et al., 2022), Transformer-alternative architectures (Gu & Dao, 2024), etc. Other works explore algorithm-level acceleration like quantization (Lin et al., 2024) or speculative decoding (Leviathan et al., 2023).

Recently, RAG-specific methods have been explored. RAG Cache (Jin et al., 2024), for example, was proposed as a multilevel dynamic caching system tailored for RAG from the system perspective. Another approach used in FiD (Izacard & Grave, 2021) and PCW (Ratner et al., 2023) parallelizes processing of individual documents and eliminates cross-document attention computations. FiD encodes each retrieved passage independently from other passages and decodes by attending over the concatenation of the resulting representations of all the retrieved passages. PCW carves a long context into chunks ("windows"), restricting the attention mechanism to apply only within each window, and re-uses the positional encodings across the windows.

Comparison with previous works that are the most relevant to our work is illustrated in Table 1. This work aims to strike an optimal balance between generation quality and computational efficiency. It is notable that the extra classifier in CRAG requires maintaining an extra model with more complex serving infrastructure; when there are $N$ contexts retrieved, there are $N + 1$ model runs in total. Our work also relies on classification to refine the retrieved documents, but it is handled by an "internal" classification process that is aligned with the generation process, so the total number of model runs in our case is $1$.

## 3 SPARSE RAG

Sparse RAG is designed for the decoder-only model architecture, which is the typical architecture of most popular LLMs. Figure 1 presents an overview of Sparse RAG inference, in which document relevance assessment is used to improve the robustness of generation. The key hypothesis of our approach is that the RAG task and per context assessment are similar tasks and the model can handle both in one shot using simple and effective training and inference techniques.

## 3.1 TRAINING PROCESS

Our work assumes that a certain amount of RAG training data–on the order of thousands of examples–is accessible, which allows us to effectively tailor and adapt existing LLMs to our specific needs. In the training phase, we integrate an additional Per Context Assessment (PCA) task into the training mixture. By incorporating the PCA task, we aim enhance the model's ability to assess the relevance of retrieved documents and respond accurately in different RAG scenarios.

**Data Augmentation with LLMs** For typical RAG data, one question-answer pair can be mapped to multiple retrieved contexts using either BM25 or an existing stand-alone retriever. However, there are cases where no golden labels indicating the quality of every retrieved context is available.

To collect these missing labels, we leveraged two off-the-shelf LLMs (Anil et al., 2023; Team et al., 2023)–PALM2 and Gemini–to assess each context. We observe empirically that a second round of prompting for critique, especially using a different model from the initial round, ensures the best quality labels. We provide our prompts in Table 11 in the Appendix. We compare different model combinations for labeling to human ground truth labels in Section 4.

**Multitasking Data Format** The LLM is trained on a mixture of two types of tasks: Per Context Assessment (relevance rating) and answer generation. Specifically, we format the inputs and outputs of the two task types as

- Per Context Assessment: $\{Question\}\{Context\}\{Control\_Assessment\}\{Rating\}$
- Generation: $\{Question\}\{Context_1\}...\{Context_N\}\{Control_Generation\}\{Answer\}$

where $\{Rating\}$ ("yes" for relevant or "no" for irrelevant) and $\{Answer\}$ are the targets for the generative tasks and all tokens before them are inputs. $\{Control\_Assessment\}$ and $\{Control\_Generation\}$ are special control tokens to ensure the LLM can differentiate the two tasks.

**Parallel Contexts** Since each context is rated independently in the PCA task, in which each example contains only one context, we introduce independence in the primary RAG generation training task as well so that the two tasks can reuse the KV cache at inference. Thus for the generation task, we enforce no cross-attention between different retrieved contexts as in Parallel Context Windows (Ratner et al., 2023).

Specifically, we modify two things in the standard LM training process. First, we change the attention masks to be block-wise, and restrict $Context_i$ and $Context_j$ from attending to one another. $\{Question\}$, $\{Context_i\}$ and $\{Control\_Generation\}$ use the default causal attention mechanism, in which the latter tokens attend to all previous ones. Second, we use "parallel incremental" positional encodings to mimic the situation in which all retrieved contexts directly follow the query while maintaining the typical position ID of the $\{Control\_Generation\}$ and $\{Answer\}$ tokens as shown below.

$$\underbrace{0, 1, 2,}_{Question} \underbrace{3, 4, 5,}_{Context_1} \underbrace{3, 4, 5, 6,}_{Context_2} \underbrace{10,}_{Control\_Generation} \underbrace{11, 12, 13}_{Answer}$$

## 3.2 INFERENCE PROCESS

Given the question and retrieved contexts, Sparse RAG handles the assessment task and generation task in one single pass.

**Per Context Assessment** Similar to the training process, when pre-filling the KV cache, each retrieved context is treated independently by masking cross-document attention. The KV cache is used to score each context by concatenating the $\{Control\_Assessment\}$ token. The relevance score is the probability of "yes" (indicating relevance) being the next token. The position encoding allows this to happen in parallel.

**Generation** The generation uses a filtered KV cache, where only $K$ out of $N$ cached values are loaded. We use a simple threshold-based filtering approach: we drop the context when its score is less than $sigma$. Once the cached KV vectors are loaded, the $\{Control\_Generation\}$ token prompts the model to generate the answer.

Table 2: Auto-rater comparison to ground truth.

| Auto-labeling method | | Average F1 | F1 Label 0 | F1 Label 1 |
| Rater model | Critic model | | | |
| --- | --- | --- | --- | --- |
| PALM2 XL | n/a | 0.729 | 0.765 | 0.694 |
| PALM2 XL | PALM2 XL | 0.781 | 0.820 | 0.741 |
| Gemini Ultra | n/a | 0.761 | 0.807 | 0.716 |
| Gemini Ultra | Gemini Ultra | 0.704 | 0.747 | 0.660 |
| PALM2 XL | Gemini Ultra | 0.728 | 0.776 | 0.680 |
| Gemini Ultra | PALM2 XL | **0.821** | **0.861** | **0.781** |

## 4 EVALUATION OF PER CONTEXT ASSESSMENT

### 4.1 NEW ANNOTATIONS: NATURAL QUESTIONS PER CONTEXT ASSESSMENT

We isolated a subset of 50 questions, each with 10 retrieved contexts, from the Natural Questions dataset. We assigned 3 raters to each question-context pair from a pool of 7 raters and provided the instructions in Section A.2.

We aggregated responses for all 3 raters for each context, selecting the majority decision 0 or 1 for each context. We found that raters unanimously agreed on 351 out of 500 context, with 30% of the documents considered relevant. For questions where raters were not unanimously decided, a specialist rater was assigned to investigate more carefully and set the best label to correct mistakes of the other raters. This resulted in 6 additional documents considered relevant out of the entire dataset, boosting the portion of relevant documents to 31% and slightly increasing alignment with the auto-rater approaches (average F-score increase of 1.4% across auto-rater methods using these corrections as the ground truth).

### 4.2 LLM RATER COMPARISONS

We tested several different LLM-based automatic labeling methods–different combinations of models and prompts–for creating training data for the classifier in Sparse RAG. We compared several of these auto-rater approaches by creating a ground-truth relevance dataset using human labeling. The auto-rater comparison using the revised human labels as the ground-truth is shown in Table 2. We find that combining two different models in two rounds–initial prompting and critique–provides the labels that are most closely aligned with the human labels. We hypothesize that the different representations learned by two different models are able to capture the most nuance in the input sequences, leading to better relevance judgements. We also observe that Gemini Ultra appears slightly less effective at critiquing model outputs than PALM2 XL.

## 5 EVALUATION OF SPARSE RAG

### 5.1 BENCHMARKS AND METRICS

**PopQA** is a large-scale open-domain question answering (QA) dataset, consisting of 14k entity-centric QA pairs. Each question is created by converting a knowledge tuple retrieved from Wikidata using a template. We follow the setup from (Yan et al., 2024) and use Contriever (Izacard et al., 2022) to retrieve the related contexts. Since PopQA does not include per-context assessment relevance labels, we adopted the "Gemini + PALM2" combination to create training labels. We split the dataset into training, validation and test sets with 8:1:1 ratio. Since the answer is usually short, we report **Exact Match (EM)** and **F1** scores.

**QMSum** (Zhong et al., 2021) is a human-annotated benchmark for a query-based multi-domain meeting summarization task, which consists of 1,808 query-summary pairs over 232 meetings in multiple domains. To adapt it to the RAG domain, we divide each conversation into different contexts where each turn in the conversation is a context and the average context contains 300 words. Note that this dataset has human labeled per-context assessments that we leverage during training. We use 250 training examples (one per meeting), 70 validation examples and 77 test examples. The targets for this dataset are longer and we report **RougeLSum** and **F1** scores.

**TriviaQA**   (Joshi et al., 2017) is a realistic text-based question answering dataset that includes 950K question-answer pairs from 662K documents collected from Wikipedia and the web. Similar to PopQA, we used the "Gemini + PALM2" combination to create relevance training labels. We randomly selected 8k training examples and 500 validation and test examples each. We report **Exact Match (EM)** and **F1** scores.

**HotpotQA**   (Yang et al., 2018) is a question answering dataset containing about 113K crowd-sourced questions that are constructed to require the introduction paragraphs of at least two Wikipedia articles to answer, thus requiring multi-hop reasoning. We sample 6k training examples and 600 validation and 600 test examples. We report **Exact Match (EM)** and **F1** scores.

These datasets were selected to demonstrate generalizability across the question answering tasks requiring single- or multi-hop reasoning, diverse context and output lengths, and summarization capabilities.

## 5.2   BASELINES

**RAG**   We evaluated the performance of standard concatenation-based RAG where an LLM generates output given the query prepended with all the top-ranked documents using the same retriever as Sparse RAG system. RAG is finetuned with the training data.

**Off-the-shelf**   We report a variant of concatenation-based RAG where the model is not finetuned with training data.

**LLMLingua**   In this approach an external LLM was called to compress the prompt (Jiang et al., 2023a). In our comparison, we chose the compression ratio to be the same as Sparse RAG for fairness.

**PCW-RAG**   We applied Parallel Context Windows (Ratner et al., 2023) to the RAG process, where no cross-attention is applied between documents. The model is finetuned with the training data.

**Corrective RAG**   We evaluate CRAG using an external T5-XXL classifier trained using heuristic labels (Yan et al., 2024). This classifier is used to process all the documents and decide the rank. Note to facilitate a fair comparison, we did not adopt the "web search" feature of this paper.

## 5.3   EXPERIMENTAL CONFIGURATION

The base LLMs used in the paper were Gemini (Team et al., 2023). Although our approach could be applied at different training stages of the model, we apply LoRA tuning (Hu et al., 2022) to enforce alignment on top of the foundation LLMs due to its low resource requirements and wide usage. Note that the same LoRA tuning on the training data is applied to Sparse RAG and all baselines. In all our experiments, we apply LoRA in self-attention and use the default rank as 4. By default, we use the XXS size of Gemini which can run on-device.

During training, we use 64 Tensor Processing Units (TPU) V3 chips for PopQA while use 128 Units for the other datasets. The batch size is 64. We use the Adafactor optimizer (Shazeer & Stern, 2018) with a learning rate of 0.003. The training dropout rate is 0.05. We leverage the metrics of the validation set to pick the best checkpoint. During inference, the temperature is set to 0.5. Unless specifically noted, we use sampling decoding with sample number 1 for our experiments.

## 5.4   INFERENCE SETUP AND METRICS

Evaluation of Sparse RAG was conducted on a Samsung S21 Ultra, utilizing the device's CPU to assess real-world performance on a relatively mid-tier smartphone compared to the latest flagship models. Inference configuration consisted of fixed token lengths for queries, contexts and generated responses. This setup allows for evaluating the system's efficiency and effectiveness under resource constraints typical of mobile devices, providing insights into its practical applicability for on-device question-answering tasks. Specifically, the overall inference process considers two stages.

**Prefill stage**   For the baseline RAG model, we measure the total time taken to process all input tokens (question and all contexts). For PCW RAG and Sparse RAG models, we take advantage of these models' ability to cache the question KV vectors. We first measure the time to process the

Table 3: Quality & efficiency tradeoff for both short-form and long-form generation tasks; Sparse RAG achieves both higher quality and efficiency compared to "dense" RAG approaches.

| Dataset | Metrics | Off-the-shelf | LLMLingua | RAG | PCW-RAG | CRAG | Sparse RAG |
|---------|---------|---------------|-----------|-----|---------|------|------------|
| - | ES | 56.28 | - | 56.28 | **147.58** | - | **147.58** |
| PopQA | EM | 0.33 | 1.96 | 65.43 | 65.04 | 66.52 | **67.71** |
| | F1 | 12.76 | 12.15 | 69.99 | 69.54 | 70.99 | **71.16** |
| | K | 20.00 | **7.84** | 20.00 | 20.00 | 8.9 | **7.84** |
| | DS | 6.65 | - | 6.65 | 6.65 | - | **12.28** |
| QMSum | F1 | 20.37 | 22.28 | 21.43 | 20.18 | - | **23.96** |
| | RougeSum | 12.67 | 18.37 | 18.20 | 16.95 | - | **20.10** |
| | K | 20.00 | **4.45** | 20.00 | 20.00 | - | **4.45** |
| | DS | 6.65 | - | 6.65 | 6.65 | - | **16.05** |
| TriviaQA | EM | 0.00 | 2.6 | 46.20 | 46.00 | - | **47.50** |
| | F1 | 12.21 | 16.30 | 53.03 | 53.20 | - | **55.10** |
| | K | 20.00 | **9.90** | 20.00 | 20.00 | - | **9.90** |
| | DS | 6.65 | - | 6.65 | 6.65 | - | **10.18** |
| HotpotQA | EM | 0.00 | 1.33 | 43 | 38.83 | - | **43.50** |
| | F1 | 12.21 | 14.67 | **55.85** | 50.03 | - | 55.36 |
| | K | 10.00 | **6.50** | 10.00 | 10.00 | - | **6.50** |
| | DS | 10.31 | - | 10.31 | 10.31 | - | **13.00** |

question alone. Then, we measure the time to process different contexts with the pre-processed KV-cached question. We use **Encoding Speed (ES)** (tokens per second (t/s)) to measure the efficiency of the prefill stage.

**Decoding stage**  To assess the decoding speed comprehensively, we generate output sequences using the same question but varying the amount of relevant context data considered, ranging from the top 1 most relevant document to the top $K$. For each context size, we produce output sequences of different lengths. This systematic approach allows us to evaluate the impact of both context size and response length on decoding speed. We use **Decoding Speed (DS)** (tokens per second (t/s)) to measure the efficiency of the decoding stage.

## 5.5 MAIN RESULTS

We report both quality and latency metrics in Table 3. "K" is the number of chosen contexts. Note that the CRAG approach relies on its classifier, which is exclusively trained on the PopQA dataset. Thus we only compare its performance on PopQA. Additionally, both LLMLingua and CRAG leveraged external classifiers, for which we cannot effectively measure ES and DS. We discuss the end-to-end latency of the external classifiers in Table 6.

Notably, our proposed approach achieves the best quality while being the most efficient during inference compared to other approaches. It can be seen that Sparse RAG shares the same pre-filling efficiency with PCW-RAG, due to the parallel context encoding, but it achieves significantly better quality than PCW-RAG amd better decoding efficiency than standard RAG and PCW-RAG. To illustrate, out of 20 retrieved contexts, Sparse RAG has an average of $7.84$ contexts for PopQA and $4.45$ contexts for QMSum. This leads to almost **double** or even **triple** the decoding speed. Meanwhile, Sparse RAG achieves higher quality metrics than the dense counterparts, demonstrating that Sparse RAG effectively filters noisy and irrelevant contexts.

HotpotQA is the only dataset where Sparse RAG does not beat RAG's F1 score. PCW-RAG has a particularly large quality gap on HotpotQA, suggesting that masking cross-document attention may hinder multi-hop reasoning capabilities. However, Sparse RAG recovers quality to a level similar to RAG while maintaining lower latency, demonstrating the power of our context selection process.

We also observe that Sparse RAG outperforms CRAG on quality, suggesting that our "in-place" classifier may be outperforming CRAG's external T5 XXL classifier trained on the same dataset.

## 5.6 ANALYSIS

**Impact of Confidence Threshold**  Table 4 illustrates how our metrics vary with different quality thresholds for Sparse RAG. As the threshold gradually increases, the system filters out more contexts,

reducing the number of contexts K and consequently the latency during inference. The response quality metrics increase with increasing threshold up to a certain point, showing the effectiveness of filtering out irrelevant contexts. Then, the performance is stable and eventually drops slightly, possibly because some relevant contexts are accidentally filtered out.

Table 4: Sampling various confidence threshold values. A higher threshold means fewer contexts.

| | PopQA | | | | QMSum | | | |
|---|---|---|---|---|---|---|---|---|
| Threshold | EM | F1 | K | DS | F1 | RougeLSum | K | DS |
| 0.05 | 66.95 | 70.97 | 9.75 | 10.61 | 22.85 | 19.49 | 7.92 | 11.92 |
| 0.1 | 66.84 | 70.66 | 8.72 | 11.70 | 23.78 | 19.98 | 6.68 | 12.89 |
| 0.15 | **67.17** | **71.16** | 7.84 | 12.28 | 23.43 | 19.66 | 5.77 | 13.01 |
| 0.2 | 66.77 | 70.54 | 7.13 | 12.88 | 23.2 | 19.79 | 5.05 | 14.54 |
| 0.25 | 65.75 | 69.64 | 6.56 | 13.00 | **23.96** | **20.1** | 4.45 | 16.05 |
| 0.3 | 63.86 | 68.2 | **5.98** | **13.08** | 23.84 | 19.99 | **3.93** | **16.38** |

**Number of Prefilled Documents**    To assess whether Sparse RAG's quality improvements are the result of "massive" prefilling which is not practical in real scenarios, we compare different numbers of prefilled documents (10 and 20) for PopQA. Results are shown in Table 5. Even with fewer documents prefilled, the quality of Sparse RAG remains better than RAG.

The gap between RAG and Sparse RAG is relatively small at 10 prefilled documents compared to 20 because there are fewer documents to be filtered. Moreover, for a smaller number of documents, the cross-document masking is less "sparse" compared to a larger number of prefilled documents. Meanwhile, only using the top 5, 3, or 1 documents introduces significantly lower EM and F1 scores because it is difficult to guarantee high-quality retrieval in the first step. This further motivates our design to increase the range of retrieved documents and then perform context selection.

Table 5: Ablation on different number of prefill documents for PopQA.

| Approach | Prefill Documents | EM | F1 | K | ES | DS |
|---|---|---|---|---|---|---|
| RAG | 1 | 146.10 | 46.01 | 50.12 | 1.00 | 22.74 |
| RAG | 3 | 120.36 | 55.28 | 59.32 | 3.00 | 18.96 |
| RAG | 5 | 102.51 | 58.66 | 63.49 | 5.00 | 15.98 |
| RAG | 10 | 64.66 | 68.67 | 10 | 80.74 | 10.31 |
| PCW RAG | 10 | 63.9 | 68.58 | 10 | 147.48 | 10.31 |
| Sparse RAG | 10 | **65.86** | **70.2** | **7.79** | **147.48** | **12.33** |
| RAG | 20 | 65.43 | 69.99 | 20 | 56.28 | 6.65 |
| PCW RAG | 20 | 65.04 | 69.95 | 20 | 147.58 | 6.65 |
| Sparse RAG | 20 | **67.17** | **71.16** | **7.84** | **147.58** | **12.28** |

**Inference Efficiency Ablations**    We present Table 6 demonstrates the computational advantages of Sparse RAG for each stage of the retrieval and generation process compared to other methods. Sparse RAG, like PCW-RAG, reduces encoding latency to nearly 1/3 that of RAG. While CRAG's encoding latency appears lower than Sparse RAG, it has an additional classification step, which is

Table 6: Latency Decomposition.

| End-To-End | K | External Classifier (ms) | Init Time (ms) | Encoding (ms) | Copy (ms) | Decoding (ms) | Total (ms) | Init + Copy Percentage |
|---|---|---|---|---|---|---|---|---|
| RAG | 20.00 | 0 | 120 | 90962 | 0 | 4811 | 95893 | 0.13% |
| PCW-RAG | 20.00 | 0 | 17 | 34716 | 151 | 4811 | 39697 | 0.43% |
| CRAG | 8.9 | 40200 | 56 | 27362 | 0 | 2878 | 70497 | 0.08% |
| Sparse RAG | 7.84 | 0 | 17 | 34716 | 56 | 2605 | 37396 | 0.20% |

slow in our experiments because we used an older T5 model with more attention layers and fewer modern optimizations like kernel fusion and flash attention. This drives CRAG's total latency above that of Sparse RAG. It is possible that a different classifier could reduce this cost, but it would still duplicate the operations of encoding the context during the prefill stage. Sparse RAG also reduces the decoding latency to nearly half that of RAG and PCW-RAG via context filtering. CRAG has similar decoding latency to that of Sparse RAG because it also uses context filtering, but overall Sparse RAG is still much faster.

We also explore trends in decoding latency by varying the number of retrieved contexts (i.e., top-K documents) and the length of the generated responses. As illustrated in Fig 2b, we observe that RAG requires over 50% more time to generate outputs of varying lengths compared to the Sparse RAG approach. As shown in Fig 2a, this heightened demand for computational resources results in a notable slowdown in decoding speed. This underscores the efficiency advantages offered by Sparse RAG, especially in scenarios requiring a larger number of contexts during decoding.

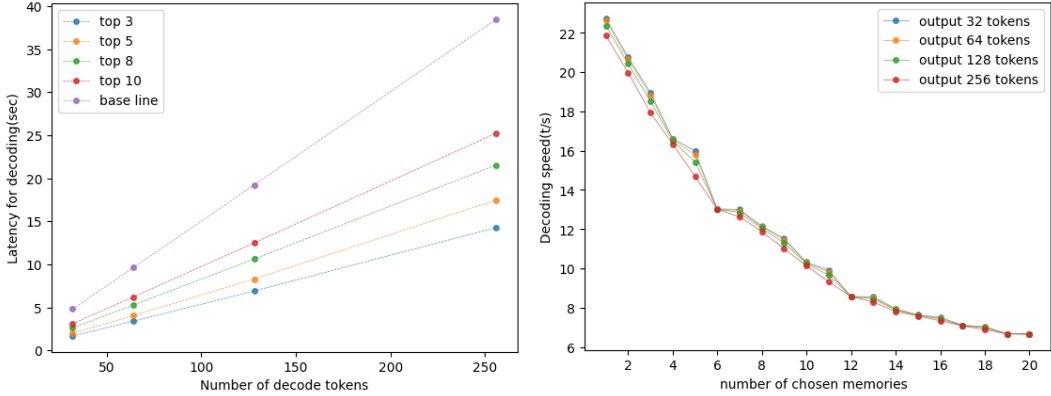

(a) E2E latency for decoding different number of tokens. (b) Decoding speed with different number of contexts.

Figure 2: Inference Efficiency Comparison.

**Ablation on Foundation Model Size**    We applied Sparse RAG to different sizes of LLMs by testing it on Gemini XS and Gemini XXS. The results of these experiments are presented in Table 7. The findings demonstrate that Sparse RAG is compatible with various foundation models, effectively adapting to different model sizes. Notably, with a reduced amount of decoding caches, Sparse RAG is capable of achieving the highest quality results. This indicates that Sparse RAG maintains its efficiency and effectiveness across different foundation models, making it a versatile approach for various LLM configurations.

Table 7: Ablation on different model sizes.

| Approach | Model Size | EM | F1 | K |
|---|---|---|---|---|
| RAG | XS | 66.52 | 70.87 | 20 |
| PCW RAG | XS | 65.75 | 70.37 | 20 |
| Sparse RAG | XS | **68.26** | **72.26** | **6.27** |
| RAG | XXS | 65.43 | 69.99 | 20 |
| PCW RAG | XXS | 65.04 | 69.95 | 20 |
| Sparse RAG | XXS | 67.17 | 71.16 | 7.84 |

**Sparse RAG with Full Attention During Generation**    To demonstrate the isolated effects of our document assessment and filtering method and to evaluate the potential quality regressions caused by omitting cross-document attention scores during generation, we perform the Sparse RAG Per Context Assessment step followed by the generation step with full attention. As you can see in Table 8, using full attention during generation provides a slight quality improvement, but Sparse RAG quality is very close and much more efficient.

Table 8: Comparing Sparse RAG with full attention to Sparse RAG and RAG on PopQA.

| Approach | EM | F1 |
|---|---|---|
| RAG | 65.43 | 69.99 |
| Sparse RAG | 67.71 | 71.16 |
| Sparse RAG w/ full attention | **67.94** | **71.24** |

**Silver Labels vs LLM Labels**    In Corrective RAG, the T5 model was trained with silver labels that come from title matching (Yan et al., 2024). We use the same silver labels to replace the LLM labels and train the Sparse RAG model with this new dataset. We also train the CRAG model on our LLM labels for comparison. From the results shown in Table 9, we observe that the quality of the labels generated by the LLMs is slightly higher than that of the silver labels from Yan et al. (2024) leading to higher accuracy and lower K values. We also observe that when trained with exactly the same labels, Sparse RAG still outperforms CRAG on quality. We hypothesize that the superior quality of the LLM-generated labels comes from our two-round process of soliciting responses from two different LLMs. By engaging two distinct models, we likely enhanced the robustness and accuracy of the labels through a form of cross-validation, thereby mitigating potential biases or errors that might arise from relying on a single LLM.

Table 9: Comparing CRAG silver labels to our LLM labels on PopQA.

| Approach | EM | F1 | K | DS |
|---|---|---|---|---|
| Sparse RAG w/ silver labels | 66.97 | 71.05 | 8.26 | 11.99 |
| Sparse RAG w/ LLM labels | **67.71** | **71.16** | **7.84** | **12.28** |
| CRAG w/ silver labels | 66.52 | 70.99 | 8.9 | - |
| CRAG w/ LLM labels | **67.03** | **71.02** | - | - |

**Using Golden Context Labels During Inference**    Since QMSum provides golden per-context labels, we leverage these labels during inference to evaluate the upper bound performance of the Sparse RAG approach under the condition of perfect per-context assessment. The results of this experiment are presented in Table 10, demonstrating the full potential of the Sparse RAG method pending additional context assessment quality improvements.

Table 10: Trying golden labels on QMSum.

| Approach | F1 | RougeLSum | K | DS |
|---|---|---|---|---|
| Sparse RAG | 23.96 | 20.1 | 4.45 | 16.05 |
| + golden label | **26.76** | **21.93** | **1.13** | **21.16** |

## 6    CONCLUSION

This paper presents Sparse RAG to address the challenges of increased input length and latency. Through a novel approach of massive pre-filling and selective decoding, Sparse RAG efficiently manages the key-value cache of retrieved documents, allowing the LLMs to focus on highly relevant tokens. This selective attention mechanism not only reduces the computational burden during inference but also enhances the generation quality by filtering out irrelevant contexts. Evaluation on four diverse datasets validates Sparse RAG's ability to achieve a balanced trade-off between high-quality generation and computational efficiency, proving its versatility and effectiveness for both short- and long-form content generation tasks. This innovative paradigm showcases the potential for improving LLM performance in various applications by optimizing context management and inference processes.

Future research will explore Sparse RAG in multimodal contexts, investigating how Sparse RAG can handle and integrate information from multiple types of data to improve its performance and applicability across diverse scenarios.

ACKNOWLEDGMENTS

We would like to thank Yu-hui Chen, Yuqi Li, Qifei Wang, Zonglin Li, Zhong Meng, and Alec Go for their suggestions.

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

## A PROMPTS AND INSTRUCTIONS

### A.1 PROMPTS USED FOR LLMS

We share the prompt used for calling LLMs to get per context assessment in Table 11.

Table 11: The zero-shot prompts for LLM labeling and critique.

| Round 1 prompt |
| --- |
| You are now doing a reading comprehension task. It is important that you be as thorough, detail-oriented, and accurate as possible in your response. |
| You are given a question, a set of accepted answers, a document and its title. The document does not necessarily contain the right answer to the question. |
| You should read the title and the document and then check if they provide one of the correct answers to the question. |
| If the title and document together contain the correct answer to the question, output a score of 1.0, otherwise output a score of 0.0. |
| question: ¡question¿ |
| accepted answers: ¡answers¿ |
| title: ¡title¿ |
| document: ¡document¿ |
| output: |

| Round 2 prompt |
| --- |
| Your job is to correct another model's performance on a reading comprehension task. |
| The model was given a question, a set of accepted answers, a document and its title. The document and title do not necessarily contain the right answer. The model was instructed to output a score of 1.0 if the document contains the answer, and a score of 0.0 otherwise. |
| You will be given the same information as the other model along with its output. You should read the title and document and then check if they provide one of the correct answers to the question. |
| Then check if you agree with the previous model's output. |
| If you agree, output the same score unchanged. |
| If you disagree, output the corrected score. |
| Your output should be as accurate as possible. |
| question: ¡question¿ |
| accepted answers: ¡answers¿ |
| title: ¡title¿ |
| document: ¡document¿ |
| previous model's score: ¡score¿ |
| output: |

### A.2 RATER GUIDELINES

We share the instructions provided to the human labelers in Table 12.

Table 12: Instructions for raters creating ground-truth relevance dataset.

| Human Rater Instructions |
| --- |
| Please read the question, the answer and the context. Please answer if the context can help answer the question. If it can, select 1. Otherwise select 0. |
| 1: good
0: bad |
| Please use the answers as a hint. However, do not use "is the answer in the context?" as a heuristic for making the decision. |

## B DATASET ANALYSIS

During the human labeling process, several raters flagged documents and questions that were difficult to label. In total, 23 out of 500 documents were flagged and 15 out of 50 questions were flagged.

We explored several ways of filtering our human-labeled subsample of Natural Questions to determine how they impacted context assessment F-scores overall and for each auto-rater. We provide two additional filtered versions of the human-labeled RAG relevance dataset as alternatives. See Table 13 for the auto-rater F-scores for each filtering method. Both statistical filtering approaches (e.g. removing contexts with non-unanimous labels) and targeted filtering approaches (e.g. removing questions or contexts flagged by human raters) lead to some improvement in F-scores for relevance labels, but in all cases, using Gemini Ultra as the rater and PALM2 XL as the critic model provides the highest Average F1 score.

Table 13: Evaluation of Labeling Methods w/ Filtering

| Dataset Filters | # Docs | % Relevant | Rater Model | Critic Model | Average F1 |
|---|---|---|---|---|---|
| Specialized rater corrections | 500 | 31 | PALM2 XL | n/a | 0.729 |
| | | | PALM2 XL | PALM2 XL | 0.781 |
| | | | Gemini Ultra | n/a | 0.761 |
| | | | Gemini Ultra | Gemini Ultra | 0.704 |
| | | | PALM2 XL | Gemini Ultra | 0.728 |
| | | | Gemini Ultra | PALM2 XL 340B | **0.821** |
| Filter non-unanimous docs | 351 | 23 | PALM2 XL | n/a | 0.741 |
| | | | PALM2 XL | PALM2 XL | 0.811 |
| | | | Gemini Ultra | n/a | 0.792 |
| | | | Gemini Ultra | Gemini Ultra | 0.739 |
| | | | PALM2 XL | Gemini Ultra | 0.763 |
| | | | Gemini Ultra | PALM2 XL | **0.856** |
| Filter flagged docs and questions | 330 | 29 | PALM2 XL | n/a | 0.750 |
| | | | PALM2 XL | PALM2 XL | 0.797 |
| | | | Gemini Ultra | n/a | 0.782 |
| | | | Gemini Ultra | Gemini Ultra | 0.741 |
| | | | PALM2 XL | Gemini Ultra | 0.753 |
| | | | Gemini Ultra | PALM2 XL | **0.833** |

We observed trends in questions and contexts shared in Table 14 that raised concerns about whether and how a human would be able to assess the relevance of the context. These concerns extend to expectations of how well LLMs would do at the task.

Most of these concerns involve the absence of sufficient context to correctly answer the question in widely used public datasets. For consistency with the literature, we did not modify the queries or the retrieved contexts in this paper, but we do expect that the ambiguity is impacting our results. In many cases, retrieved contexts would be assessed differently for relevance depending on the true intended meaning of the question, and different answers would be expected.

It would be an interesting future exploration to augment the datasets to resolve such ambiguity, as it would likely improve the accuracy of our relevance labels and improve performance overall. For examples with missing time or location context, you could augment the datasets by simply adding new contexts with the missing information or appending the context to the query. For example, to handle the question "Who is the current president of the United States?", you could add a context "Today is November 18, 2024." or you could modify the query to "Who is the current president of the United States on November 18, 2024"?. In the paper, we showed that Sparse RAG performs comparably to RAG on the HotpotQA multi-hop reasoning dataset, so we expect that Sparse RAG could effectively leverage information from new contexts in the generation step. However, we expect that appending such temporally specific context to the query itself would likely yield the best quality, as all contexts attend to the query.

In real world cases where the question is ambiguous and sufficient context cannot be retrieved to confidently answer the question, we believe that the most desirable behavior would be to request clarification before further action. Future work should explore simulating such a scenario in order to assess RAG question-answering approaches.

Table 14: Overview of trends, datasets, and examples with associated comments.

| Trend | Dataset | Example question | Comments |
|---|---|---|---|
| Questions with time-dependent answers | NQ | who is the president of usa right now | Depends on when the question is asked. |
| | NQ | who is the current director of the us mint | Depends on when the question is asked. |
| | NQ | when is the next deadpool movie being released | Depends on when the question is asked. |
| | NQ | total number of death row inmates in the us | Fluctuates over time. |
| | PopQA | What is Prague the capital of? | The borders in this region and the name of the country have changed several times in the 20th century. |
| | PopQA | What is Dennis Rodman's occupation? | The accepted answers are "actor, actress, actors, actresses". He was an actor later in his career, but he rose to prominence as a professional basketball player. |
| Missing synonyms, abbreviations, and aliases, combined with unclear granularity | NQ | in which regions are most of Africa petroleum and natural gas found | "Region" can refer to different levels of granularity (e.g. Sub-Saharan Africa vs. Ethiopia), but the only accepted answer is "Nigeria". |
| | NQ | what type of car is a jeep | The accepted answers are only "off-road vehicles", "light utility vehicles", "sport utility vehicles", but "SUV" is clearly a correct answer as well. |
| Non-exhaustive list of answers | NQ | cast of law & order special victim unit | The accepted answers include 16 cast members, but the show went on for 25 seasons with many cast changes and guest stars not included in the list. |
| Oddly phrased question | NQ | right to property according to the constitution of India is a | The only correct answer is "constitutional right", but that is included in the question. It's not clear what type of answer would be appropriate here. |
| Overly specific answers expected | NQ | where does the story the great gatsby take place | The only accepted answer here is "Long Island of 1922", but the place is Long Island and the question does not ask about when the story is set. |
| Question refers to an entity with a common name without disambiguation | PopQA | What genre is Frances? | There is a musician and a film called "Frances", and both of those could arguably have a genre associated with them. |
| | PopQA | Who was the producer of Hurt? | The question is referring to a song performed by Christina Aguilera but there are many other songs, movies, and other entities that share the name and also have a producer. |
| | PopQA | What is the capital of Cherokee County? | There are many different Cherokee Counties in different states in the USA. |