# OpenReview forum: "Accelerating Inference of Retrieval-Augmented Generation via Sparse Context Selection"
_ICLR.cc/2025/Conference — ICLR 2025 Poster_

### Official Review · Reviewer_pGev · 2024-11-01

**Soundness:** 3
**Presentation:** 3
**Contribution:** 2
**Rating:** 5
**Confidence:** 4

**Summary:**

This work proposes Sparse RAG, a novel RAG framework that reduces computational costs by leveraging sparsity. Sparse RAG encodes retrieved documents in parallel and allows LLMs to selectively decode by attending only to the most relevant cached information. Experiments on four datasets demonstrate that Sparse RAG effectively balances generation quality with computational efficiency.

**Strengths:**

1. The proposed framework effectively balances generation quality with computational efficiency.

**Weaknesses:**

1. This work lacks novelty; the authors' primary contribution lies in scoring documents and using an existing framework with modified position encodings to achieve parallel encoding.
2. The experiments lack ablation studies for the module, especially an experiment where only Parallel Contexts is added to RAG (for only generation task).
3. Some parts of this work are not clearly written, making them difficult to understand. For example, lines 201 to 204, with line 204 being especially unclear.

**Questions:**

1. What is the detailed difference between Parallel Context Windows (PCW) and the Parallel Contexts the authors proposed? A specific example would be helpful.
2. In line 203, what is the {Relevant} token?
3. In Figure 1, do $c_{i}$ and $d_{i}$ represent the same thing? If not, what’s the difference between them?
4. Is Parallel Contexts used for both assessment and generation? The authors state that "we unify the generation of the special control tokens and regular vocabulary tokens with one single model," but line 188 mentions that Parallel Contexts is used only for the "generation task." Does this represent a contradiction?
5. In the section *Silver Labels vs. LLM Labels*, could an experiment be added to evaluate Corrective RAG using LLM Labels? The improvement shown in Table 5 appears minimal, which may be influenced by controllable or uncontrollable factors.

---

> ### Author Response · Authors · 2024-11-19
>
> First of all, we would like to apologize that we did not present our work clearly enough in our paper.  For the questions below, we will update our manuscript to make sure things are clear.
>
> >   Primary contribution lies in scoring documents and using an existing framework with modified position encodings to achieve parallel encoding
>
> We think this understanding is not accurate. Our novelty not only lies in combining "parallel encoding" with "scoring", but also in demonstrating that we can perform the two separate tasks in just one step with one model without any additional cost. This is significantly different from previous works which either only focus on parallel contexts or use external models to filter the context. To the best of our knowledge, this is novel and no other works have explored a similar approach. Our significant gains in both quality and efficiency in Table 3 should help justify our novelty.
>
> > The experiments lack ablation studies for the module, especially an experiment where only Parallel Contexts is added to RAG (for only generation task)
>
> We have this setup as a baseline, which is PCW-RAG.
>
> We have a number of ablations included in the paper, and we are adding an ablation in this rebuttal confirming that Sparse RAG without inter-document attention scores performs almost as well as Sparse RAG with full attention during the generation step, and both perform better than RAG.
>
> PopQA|EM|F1
> ---|:---:|---:
> SparseRAG|67.71|71.16
> SparseRAG + full attention|67.94|71.24
> RAG|65.43|69.99
>
> > Some parts of this work are not clearly written. lines 201 to 204, with line 204 being especially unclear.
>
> We apologize for the confusion. We will update our manuscript to be more clear.
>
> The LLM input format is `<Question> <Document> <Control_Assessment>`, and output format is
> `<Relevant>`, where `<Control_Assessment>` is a special control token used to instruct the LLM to generate a relevance score, and `<Relevant>` is a token either “yes” (meaning it is relevant) or “no” (meaning it is not relevant). During inference time, we add “yes” as the `<Relevant>` token so that the LLM generates the probability that the document is relevant to the current context. This probability is used as the relevance score to filter the documents.
>
> Please let us know if this is still not clear. We are happy to further clarify.
>
> > What is the detailed difference between Parallel Context Windows (PCW) and the Parallel Contexts the authors proposed?
>
> There is no difference. We adopted this idea in our approach. But we added the Per Context Assessment and context filtering using the same model to address PCW's limitations that 1/ it creates a quality regression and 2/ it only speeds up the prefill stage, not decoding. Our method, on the other hand, addresses both problems.
>
> > what is the {Relevant} token
>
> It is “yes” and “no” which defines relevant in a binary fashion. We will update this in the manuscript.
>
> > do ci and di represent the same thing?
>
> No. Di is the retrieved documents. Ci is the context KV cache of the documents.
>
> > Is Parallel Contexts used for both assessment and generation?
>
> Yes it is. When we say “Parallel Contexts is used only for the generation task.", we refer to the assessment training task, which only has 1 context, but in both training and inference, in both prefill and generation, we apply a parallel contexts mask. We will clarify this in our new manuscript.
>
> > experiment to evaluate Corrective RAG using LLM Labels?
>
> Thanks for this great suggestion. We added this ablation in the table below. The results show that if we adopt our LLM labels with CRAG we see marginal improvements in performance -- but the results still lag behind our approach. Sparse RAG still outperforms CRAG on both quality and efficiency.
>
> PopQA | EM | F1
> ---|:---:|---:
> CRAG | 66.52 | 70.99
> SparseRAG | 67.71 | 71.16
> CRAG (LLM Labels) | 67.03 | 71.02

---

> > ### Comment · Reviewer_pGev · 2024-11-23
> >
> > Thank you for your responses. They have addressed most of my concerns and made this work comprehensive. Although there is still some lack of novelty, I will raise my score to reflect the enhanced quality of your work.

---

### Official Review · Reviewer_4HHs · 2024-11-03

**Soundness:** 3
**Presentation:** 3
**Contribution:** 3
**Rating:** 6
**Confidence:** 3

**Summary:**

This paper proposes SparseRAG, a method to improve efficiency/effectiveness and reduce computation costs through sparsity. It encodes retrieved documents in parallel, which eliminates latency introduced by long-range attention of retrieved documents. Then, LLMs selectively decode the output by only attending to highly relevant caches auto-regressively.  This is via prompting LLMs with special control tokens. Sparse RAG can facilitate the reduction of the number of documents loaded during decoding for accelerating the inference of the RAG system. It filters out undesirable contexts and enhances the model’s focus on relevant context, improving its generation quality. Evaluations on four datasets show good results on improving generation quality and computational efficiency across multiple tasks.

**Strengths:**

1.	Overall, the paper is clearly written.

2.	The motivation of SPARSE RAG is natural, and the design of the approach seems to be innovative and reasonable.

3.	The claim that “Sparse RAG combines the assessment of each individual context and the generation of the response into a single process, in which special control tokens are used to prompt the LLM to assess the relevance of each retrieved context, and then only the key-value caches of the most relevant contexts are loaded for decoding using another control token.”  This seems to be interesting and novel, but some detailed explanation is needed to help readers understand the process.  For example, it will be good to explain how the special control tokens work in practice, or to provide a step-by-step explanation of how the context assessment and response generation are combined.

**Weaknesses:**

1.	The design assumes that a good amount of RAG training data is accessible, which allows effective tailoring and adapting of existing LLMs to the specific needs. But this design relies on pretty heavy training data, for example, in the PoPQA testing, the dataset is split into training, validation and test sets with 8:1:1 ratio (and other testing follows a similar spirit).  I am wondering in practice, an LLM will take many kinds of queries and refer to different kinds of external data, where the training pairs may not be available.  How does your method work if such training data is not available, or when your data and applications do not match the statistics in your training data?  I would suggest that the authors conduct additional experiments with varying amounts of training data to demonstrate the method's robustness. Additionally, it will be useful to discuss potential strategies for adapting the method to domains where extensive training data is not available.

2.	The design does not do any concrete semantic analysis on query intent and nor does any concrete semantic analysis on the text to be retrieved.  It would be helpful to discuss the potential benefits and drawbacks of incorporating semantic analysis into this approach. The reviewer is wondering whether the authors should discuss the potential effect of semantic analysis of the queries and texts and why they chose not to include them in their current design.

**Questions:**

The points outlined in the “weakness” should be clarified in the rebuttal.

Also, in your appendix, you mentioned "We observed trends in questions and contexts shared in Table 12 that raised concerns about whether and how a human would be able to assess the relevance of the context. These concerns extend to expectations of how well LLMs would be able to do that as well."

For example, you mentioned that in the dataset NQ, a query like "who is the president of USA right now", and you commented that "the answer will depend on when the question is asked".

This is true but I assume that is exactly why RAG could be useful since RAG will retrieve additional information based on the context of the query, and in many cases, information in the query, its surrounding context and the supporting datasets should be able to disambiguate the time period and thus return the correct answer.   I think some discussion on this is necessary.  It would be helpful if the authors may elaborate on how the Sparse RAG approach specifically handles such time-sensitive or ambiguous queries. The authors may include examples or experiments demonstrating how their method performs on queries that require temporal context, or a discussion on how such capabilities can be incorporated in the Sparse RAG framework.

---

> ### Author Response · Authors · 2024-11-19
>
> >This design relies on pretty heavy training data. How does your method work if such training data is not available, or when your data and applications do not match the statistics in your training data?
>
> In this paper, we proposed a solution to this challenge. Specifically, we leveraged LLM critiques to generate synthetic data when suitable training data was unavailable. As demonstrated in our evaluation, this approach proved to be highly effective, outperforming silver-label methods in the case of PopQA. While testing in out-of-distribution scenarios is not the primary focus of this study, we emphasize the potential of employing LLM critiques more effectively, given their demonstrated ability to assess document relevance with notable accuracy [1]. Additionally, by collecting as much available data as possible and employing ensemble techniques, we can derive a distribution that is more generalizable across domains and datasets lacking sufficient training data.
>
> [1] Retrieval Augmented Generation or Long-Context LLMs? A Comprehensive Study and Hybrid Approach. EMNLP 2024.
>
> > semantic analysis on query intent and nor does any concrete semantic analysis on the text to be retrieved.
>
> Our design intentionally avoids performing explicit semantic analysis of query intent and the text to be retrieved to prioritize simplicity and computational efficiency. We acknowledge that incorporating semantic analysis could potentially enhance the retrieval accuracy, especially in cases where query intents or ambiguous language need to be interpreted. Such analysis might better capture subtle relationships between queries and documents, leading to more precise matching. However, there are also potential drawbacks to integrating semantic analysis. It could significantly increase the system’s complexity and computational overhead, potentially impacting scalability.
>
> In this study, we chose not to include explicit semantic analysis in our design because our primary focus was on evaluating the effectiveness of leveraging LLM critiques in achieving a balance between the efficiency of RAG inference. We aimed to establish a baseline performance that could later be extended or refined in future iterations. Nevertheless, we agree that discussing this tradeoff would provide valuable context, and we will include a discussion on the potential impact of semantic analysis and the rationale behind our design choices.
>
> >  elaborate on how the Sparse RAG approach specifically handles such time-sensitive or ambiguous queries.
>
> Thanks for this question. Section B in the Appendix addresses the absence of sufficient context to correctly answer the question in the widely used public datasets we experimented on. For consistency with the literature, we did not modify the queries or the retrieved contexts, but we do expect that the ambiguity is impacting our results. In many cases, retrieved contexts would be assessed differently for relevance depending on the true intended meaning of the question, and different answers would be expected. It would be an interesting future exploration to augment the datasets to resolve such ambiguity, as it would likely improve the accuracy of our relevance labels and improve performance overall.
>
> For missing time or location context, you could augment the datasets by simply adding new contexts with the missing information or appending the context to the query. For example, to handle the question "Who is the current president of the United States?", you could add a context "Today is November 18, 2024" or you could modify the query to "Who is the current president of the United States on November 18, 2024"?. In the paper, we showed that Sparse RAG performs comparably to RAG on the HotpotQA multi-hop reasoning dataset, so we expect that Sparse RAG could effectively leverage information from new contexts in the generation step, but appending such temporally specific context to the query itself would likely yield the best quality, as all contexts attend to the query.
>
> In real world cases where the question is ambiguous, the most desirable behavior would be to request clarification before further action. In an experimental environment, we could use LLMs to rewrite the queries to contain disambiguating details without giving away the answers. Alternatively, one could simulate requesting clarification by adding an additional binary ambiguous/clear classification task for each question and augmenting the dataset with disambiguating information similar to the approach described above. Then, in evaluation, the model would classify the question's ambiguity, then request more information if necessary, then perform per-context assessment, and then perform generation.
>
> We will add some discussion of this to the new manuscript Appendix.

---

> > ### Comment · Reviewer_4HHs · 2024-11-26
> >
> > I think the author answered my concerns satisfactorily based on authors' positions.  I think the paper does make a contribution to the current state-of-the-art.   However, I think an approach that does not address semantic analysis on the text/queries and not address ambiguity in queries by semantic analysis will still impose serious limitations on the effectiveness of retrieval.   Thus, I will maintain my original relatively positive evaluation score on this paper.

---

> ### Comment · Area_Chair_MBdp · 2024-11-25
> **Reminder: Rebuttal Deadline for ICLR 2025**
>
> Dear Reviewer 4HHs,
>
> As the rebuttal deadline approaches, please kindly check the papers' discussion threads and respond to the authors' rebuttals. If you haven't had a chance to respond yet, I’d greatly appreciate your input soon. Your insights are invaluable to the authors and the review process.
>
> Thank you for your effort and support!
>
> Best regards,
>
> Area chair

---

### Official Review · Reviewer_5W8k · 2024-11-03

**Soundness:** 3
**Presentation:** 3
**Contribution:** 3
**Rating:** 6
**Confidence:** 3

**Summary:**

The paper proposes "Sparse Retrieval-Augmented Generation" (Sparse RAG), a paradigm to accelerate large language model (LLM) inference by selectively loading only the most relevant retrieved contexts. This approach aims to improve computational efficiency and generation quality by filtering unnecessary documents during decoding, evaluated across diverse datasets with positive outcomes in both quality and speed.

**Strengths:**

1. Proposes an innovative sparsity-driven retrieval method (Sparse RAG) that enhances both inference efficiency and output quality.
2. Successfully demonstrates generalizability and adaptability through evaluations on multiple tasks and datasets.
3. Offers thorough comparative analysis against existing methods, providing clear evidence of computational and quality improvements.
4. Utilizes a practical on-device evaluation setting to substantiate real-world applicability, particularly on resource-constrained devices.

**Weaknesses:**

1. The paper does not provide experimental comparisons to validate that the absence of inter-document attention scores during inference does not lead to a significant decline in the quality of responses. This aspect could be further explored to strengthen the claims made regarding the effectiveness of the Sparse RAG approach.
2. The training process involves multi-tasking and parallel context encoding, which may introduce additional complexity and require careful tuning to ensure effectiveness.

**Questions:**

1. Does reusing the KV Cache during batch inference cause a decline in the quality of responses when the document-level attention scores are missing?
2. Why does the use of an external classifier in the CRAG method increase so much latency? (My understanding is that the classifier should be a model with relatively few parameters, and it should be able to perform batch inference.) Is the training data for the classifier consistent with the PCA task?

**Details Of Ethics Concerns:**

Thank you for the input. I've made the appropriate changes based on your response.

---

> ### Author Response · Authors · 2024-11-19
>
> > The paper does not provide experimental comparisons to validate that the absence of inter-document attention scores during inference does not lead to a significant decline in the quality of responses.
>
> We would like to clarify that our experiments already show that the absence of inter-document attention scores during inference does lead to a quality regression. PCW-RAG, which is one of our baselines, uses inter-document attention masking without our document assessment training and filtering. In Table 3 in the paper, PCW-RAG has a clear drop in quality, especially for HotpotQA which requires multi-hop reasoning. Sparse RAG, on the other hand, which also omits inter-document attention during inference, remedies this quality drop by removing irrelevant contexts from the generation step, achieving better quality than RAG.
>
> Moreover, as part of this rebuttal, we added an additional ablation confirming that the Sparse RAG without inter-document attention scores performs almost as well as Sparse RAG with full attention during the generation step, and both perform better than RAG.
>
> PopQA|EM|F1
> ---|:---:|---:
> SparseRAG|67.71|71.16
> SparseRAG + full attention|67.94|71.24
> RAG|65.43|69.99
>
> > Additional complexity and require careful tuning to ensure effectiveness.
>
> Although our approach does add some additional complexity and does require further training, the changes required are minimal and only affect the position encoding and the attention mask. This is little work to implement and adds little additional complexity in terms of training efficiency. More importantly, our approach minimizes computation and requires fewer resources during inference time, which could be seen as a complexity decrease.
>
> Regarding “careful tuning”, our experiments show that a natural combination of the two training tasks–including all retrieved contexts as examples for the classification task and all queries for the generation task–works well. Adjusting the mixture weight does not cause obvious differences in performance. This suggests that tuning is pretty stable.
>
> > Does reusing the KV Cache during batch inference cause a decline in the quality of responses when the document-level attention scores are missing?
>
> Yes, it does, but as we explained in our response to your first point above, the document assessment step improves quality beyond that of full-attention RAG with large efficiency gains.
>
> > Why does the use of an external classifier in the CRAG method increase so much latency?
>
> The T5 model compared here is a T5 large, which is a 750M size model. On the contrary, the LLM profiled here is the XXS size of Gemini (scale of 1000M), which can run on-device. Thus, this classifier is not small in our settings. Also, note that our baseline model Gemini has a different implementation from the old-style LLM (T5). For example, T5 has a much larger number of attention layers than the XXS version of Gemini. Gemini XXS is also implemented with flash attention and other optimizations which improve speed, as opposed to T5’s model, which does not include these performance optimizations.
>
> > Is the training data for the classifier consistent with the PCA task
>
> Yes, the classifier is also trained with the PopQA dataset so that the methods can be fairly compared.

---

> ### Comment · Area_Chair_MBdp · 2024-11-25
> **Reminder: Rebuttal Deadline for ICLR 2025**
>
> Dear Reviewer 5W8k,
>
> As the rebuttal deadline approaches, please kindly check the papers' discussion threads and respond to the authors' rebuttals. If you haven't had a chance to respond yet, I’d greatly appreciate your input soon. Your insights are invaluable to the authors and the review process.
>
> Thank you for your effort and support!
>
> Best regards,
>
> Area chair

---

> > ### Author Response · Authors · 2024-12-02
> >
> > Dear Reviewer 5W8k,
> >
> > This is a friendly reminder that the window for reviewer responses closes today. We are very interested to hear your your thoughts on our rebuttal, and we would greatly appreciate it if you can find a few minutes to respond today.
> >
> > Thanks again for your time!
> >
> > Best,
> > The Authors

---

### Official Review · Reviewer_ARp8 · 2024-11-03

**Soundness:** 3
**Presentation:** 3
**Contribution:** 2
**Rating:** 8
**Confidence:** 4

**Summary:**

This paper presents SparseRAG, an efficient framework designed to address the scaling challenges of traditional RAG models. The primary issue tackled is the significant increase in memory usage of the key-value (KV) cache when processing large amounts of retrieved lengthy documents. This issue hurts the ability of large language models (LLMs) to efficiently manage long-form context and capture essential content.

To overcome this, the authors propose SparseRAG, which utilizes a required LLM rater and an optional LLM critic to generate Per-Context-Assessment (PCA) training data. The LLM is then trained on this PCA data to enhance its capability in assessing the relevance of retrieved documents. During both training and inference, after retrieving the related contexts, the LLM first evaluates the relevance of each retrieved documents and simultaneously extracts the KV caches, which are then concatenated to generate the final outputs.

The experimental results indicate that SparseRAG significantly improves both prefill and decoding efficiency. Additionally, it outperforms all other baseline models in generating high-quality outputs. Further analysis and ablation studies confirm that SparseRAG is both an efficient and effective framework.

**Strengths:**

1. The writing is clear and easy to follow, addressing a genuine concern.

2. The proposed method demonstrates efficiency and effectiveness compared to the baselines.

3. Most ablation studies and analyses are convincing.

**Weaknesses:**

1. Ablation Study Suggestion: It would be beneficial to include an ablation study to evaluate the impact of concatenating selected KV caches instead of the corresponding raw contexts. This approach aims to save computation time, but it’s important to assess any potential performance loss compared to recalculating KVs with full attention on concatenated raw contexts. Understanding this trade-off is crucial, as certain applications might accept additional computation time if it results in significant improvements.

2. Comparison with KV Cache Compression Methods: Since the paper claims that SparseRAG enhances efficiency during the prefill and decoding stages by adopting context selection, it would be compelling to compare it with KV cache compression methods such as H2O, SnapKV, and KVMerger. Although these are typically training-free methods, and thus challenging to directly compare, such an analysis could provide valuable insights into the relative benefits and limitations of SparseRAG.

**Questions:**

See the weakness part.

---

> ### Author Response · Authors · 2024-11-19
>
> > ablation study to evaluate the impact of concatenating selected KV caches instead of the corresponding raw contexts
>
> Thanks for this great suggestion. We share results on the PopQA dataset in the table below, where we used a full-attention model to process the same selected context. Using full attention provides a slight quality improvement, but Sparse RAG quality is very close and more efficient.
>
> PopQA|EM|F1
> ---|:---:|---:
> SparseRAG|67.71|71.16
> SparseRAG + full attention|67.94|71.24
> RAG|65.43|69.99
>
> > Comparison with KV Cache Compression Methods
>
> Thank you for the thoughtful suggestion. We acknowledge the potential insights that such a comparison could provide. However, our primary focus in this work is to demonstrate the effectiveness of SparseRAG’s context selection strategy in enhancing efficiency during prefill and decoding stages, which we believe represents a substantial and novel contribution in itself.
> While KV cache compression methods typically operate under different paradigms and are training-free, integrating a detailed comparison would require addressing additional factors such as varying use cases and performance metrics. Conducting such an analysis is a non-trivial extension that we believe is better suited as a direction for future research.
>
> We appreciate your suggestion, and it will certainly inspire further exploration of these ideas in subsequent work. For now, we hope the presented results of comparing LLMLingua and PCW-RAG sufficiently illustrate the utility and advantages of SparseRAG in its intended scope.

---

> > ### Comment · Reviewer_ARp8 · 2024-11-25
> > **Response to the ablation study results.**
> >
> > Thanks for the results. I've already made necessary changes.

---

### Official Review · Reviewer_p444 · 2024-11-06

**Soundness:** 3
**Presentation:** 3
**Contribution:** 3
**Rating:** 8
**Confidence:** 4

**Summary:**

The paper proposes a sparse RAG which consists of two stages – 1) document assessment, and 2) generation, to improve the efficiency of RAG. The document assessment selects the relevant documents by individually checking their relevance, and generation part only uses the selected documents as a context to generate an answer, but in an efficient manner where documents are cached, and cross-attention b/w documents is not allowed. To my understanding, the efficiency is obtained here, because all previous information could be cached (no recomputation) after document assessment. Experiment results show that a sparse RAG leads to improvements in terms of both effectiveness and efficiency, i.e., it improves RAG, more efficiently than PCW RAG.

**Strengths:**

1.	The paper is clearly written and the proposed sparse RAG by appending document assessment stage is well validated.
2.	In the experiments, the proposed method leads to additional improvements in terms of both RAG and sparse RAG. It is more efficient than RAG, and it is remarkable the sparse RAG shows improved performances over RAG.

**Weaknesses:**

1.	Some parts in the presentation are not clear. In multitasking data format, more details of the format need to be provided. E.g. What is the information of Rating field? In Figure 1, [ASSESS] and [GENERATE] are special tokens? Detailed position information for cached document representations used in document assessment and generation. Document assessment does not use the cached representations in input?
2.	Context reduction methods have been widely studied for RAG. Document assessment method needs to be validated separately. Other context compression and reduction baselines (not only RAG) need to be compared.
3.	Given the restricted cross-attention manner across passages, there is concern that the general ability is still well-maintained. SparseRAG needs to be further validated under other advanced CoT, long chain of reasoning, and instruction following tasks.
4.	Document assessment can also be parallelizable, using restricted cross-attention method? Still, it remains unclear whether the current decision of relevance for documents in an one-by-one manner is optimally designed.

**Questions:**

Please see weaknesses.

---

> ### Author Response · Authors · 2024-11-19
>
> > Some parts in the presentation are not clear
>
> We apologize for any unclear presentation. We will provide clarification in the new manuscript. To address your questions, 1/ The rating tokens are “yes” and “no”.  2/ Yes, “assess” and “generate” are special tokens. 3/ The position encoding is "parallel incremental”. For example, in “0 1 2 3 4 5 3 4 5 6 10 11 12”,  (0 1 2) is the question, (3 4 5) is the first context, (3 4 5 6) is the second context, and (10 11 12) is the response. 4/ During inference, the assessment does use the cached representations in input.
>
> > Context reduction methods have been widely studied for RAG
>
> Thanks for this suggestion. We understand the importance of comprehensive comparisons to situate our work within the broader landscape of research. We would like to highlight that our baselines already include two effective and popular approaches specifically focused on context reduction, i.e., CRAG and LLMLingua. These baselines were selected due to their relevance and strong performance in addressing the challenges of context compression. By including these methods, we ensure a robust and meaningful evaluation of our proposed approach. While we agree that more baselines could theoretically provide further insights, we believe that our current selection is sufficient to demonstrate the effectiveness of our method, particularly given the computational and practical constraints of extensive comparative studies. We remain open to expanding this comparison in future work as additional SOTA methods emerge or if a specific context warrants further exploration.
>
> > Document assessment method needs to be validated separately
>
> We have validated our document assessment method separately in several ways. 1/ We created a golden dataset with 3 human annotators on a subsample of the Natural Questions dataset and compared F1-score for different LLM labeling methods to select the best one. Details are in Section 4 and the Appendix. 2/ We performed an end-to-end evaluation of our label quality compared to silver labels from the CRAG classifier training data (derived heuristically). We showed that all quality and efficiency metrics improved by using our Sparse RAG LLM-generated labels.
>
> We also add an ablation in this rebuttal using our document assessment and filtering method followed by generation with full attention. It shows the isolated effects of our document assessment method. The method provides quality improvements over standard RAG with and without cross-document attention. We are open to performing further assessment if you have specific requests.
>
> PopQA|EM|F1
> ---|:---:|---:
> SparseRAG|67.71|71.16
> SparseRAG + full attention|67.94|71.24
> RAG|65.43|69.99
>
> > SparseRAG needs to be further validated under other advanced CoT, long chain of reasoning, and instruction following tasks.
>
> Thanks for this suggestion. Our proposed method can be considered an instantiation of CoT. Specifically, Sparse RAG first determines the relevance of each document to the query. Then, it performs filtering based on the document assessment result. Finally, it utilizes the document caches for generation. This CoT-inspired process leverages the reasoning capabilities of LLMs. Experimentally, we evaluated Sparse RAG on the HotpotQA dataset, which requires multi-hop CoT reasoning. The results demonstrate that Sparse RAG achieves comparable quality to RAG while offering significantly improved efficiency.
>
> > Document assessment can also be parallelizable, using restricted cross-attention methods?
>
> We would like to clarify that our document assessment is not performed one-by-one; it is already parallelizable using restricted cross-attention because documents are treated independently by masking inter-document attention. This improves efficiency for both server-side and on-device inference.

---

> ### Comment · Area_Chair_MBdp · 2024-11-25
> **Reminder: Rebuttal Deadline for ICLR 2025**
>
> Dear Reviewer p444,
>
> As the rebuttal deadline approaches, please kindly check the papers' discussion threads and respond to the authors' rebuttals. If you haven't had a chance to respond yet, I’d greatly appreciate your input soon. Your insights are invaluable to the authors and the review process.
>
> Thank you for your effort and support!
>
> Best regards,
>
> Area chair

---

### Meta-Review · Area_Chair_MBdp · 2024-12-19

**Metareview:**

Summary of the paper: This paper introduces Sparse RAG designed to enhance the efficiency and effectiveness of traditional RAG. Sparse RAG addresses the challenges associated with increased memory usage of KV caches when processing lengthy documents, which can impede LLMs in managing long-form contexts effectively. Sparse RAG operates in two main stages: document assessment and generation. In the document assessment phase, relevant documents are evaluated for relevance using a required LLM rater and an optional LLM critic, generating PCA training data. During training and inference, the LLM assesses the relevance of retrieved documents, extracting and caching KV information for efficient processing. By encoding retrieved documents in parallel, Sparse RAG eliminates the latency associated with long-range attention mechanisms, allowing LLMs to decode outputs by selectively attending to only the most relevant cached information. This approach, facilitated by special control tokens, improves computational efficiency and generation quality by filtering out unnecessary contexts. Experimental results show that Sparse RAG significantly enhances both prefill and decoding efficiency while outperforming baselines.

Strengths of the paper:
- Clear Presentation: The paper is generally well-written and easy to follow, but some sections require further clarification to enhance readability.
- Well-motivated Method: The motivation of Sparse RAG is natural, and the design of the approach is innovative and reasonable.
- Good Empirical Results: Sparse RAG is efficient and effective compared to baselines across multiple tasks and datasets. The authors provide a thorough comparative analysis against existing methods, offering clear evidence of computational and quality improvements, and emphasizing real-world applicability through practical on-device evaluations.

Weaknesses of the paper:
- Missing Ablation Studies (Reviewers p444, ARp8, 5W8k): Include an ablation study to evaluate the impact of concatenating selected KV caches instead of the corresponding raw contexts.
- Needs Further Dicussions on Specific Queries (Reviewer 4HHs): Elaborate on how the Sparse RAG approach specifically handles such time-sensitive or ambiguous queries.
- Missing Comparisons with Corrective RAG (Reviewer pGev): Include experiment to evaluate Corrective RAG using LLM Labels.

Reasons for the decision:
After considering the rebuttal, I believe the authors have adequately addressed most of the concerns raised. The reviewers discussed these points in detail, and all agree that the paper should be accepted at ICLR. Reviewer pGev rated the paper a 5, which is marginally below the acceptance threshold; however, after further discussion with the authors, they expressed their belief that this work is worthy of acceptance. Overall, Sparse RAG effectively balances generation quality with computational efficiency, marking a promising advancement in the field of retrieval-augmented generation systems.

**Additional Comments On Reviewer Discussion:**

Most of the concerns raised have been adequately addressed by the authors:
- Missing Ablation Studies (Reviewers p444, ARp8, 5W8k):  The authors provide results on the PopQA dataset, where a full-attention model is used to process the same selected context. Using full attention provides a slight quality improvement, but Sparse RAG quality is very close and more efficient.
- Needs Further Discussions on Specific Queries (Reviewer 4HHs): The authors provide the discussion and promise to attach it in the appendix.
- Missing Comparisons with Corrective RAG (Reviewer pGev): The authors experiment to evaluate Corrective RAG using LLM Labels.  LLM labels with CRAG still lag behind Sparse RAG in terms of quality and efficiency.

---

### Decision · Program_Chairs · 2025-01-22

Accept (Poster)